A turiasaurian sauropod dinosaur from the Early Cretaceous Wealden Supergroup of the United Kingdom

Mannion Philip D. philipdmannion@gmail.com
Department of Earth Science and Engineering, Imperial College London , London , UK
Sues Hans-Dieter
Electronic publication date: 2019 Jan 24
Publication date: 2019
Volume: 7
Electronic Location ID: e6348
Received 2018 Oct 8; Accepted 2018 Dec 27
Copyright: © 2019 Mannion
Copyright year: 2019
Copyright holder: Mannion
License: This is an open access article distributed under the terms of the Creative Commons Attribution License, which permits unrestricted use, distribution, reproduction and adaptation in any medium and for any purpose provided that it is properly attributed. For attribution, the original author(s), title, publication source (PeerJ) and either DOI or URL of the article must be cited.
License URL: https://creativecommons.org/licenses/by/4.0/

Keywords: Cretaceous, Eusauropoda, Wealden, Biogeography, Turiasauria, Jurassic/Cretaceous boundary, Laurasia, Mesozoic

Funding: Royal Society University Research Fellowship UF160216 This research was supported by a Royal Society University Research Fellowship (UF160216). The funders had no role in study design, data collection and analysis, decision to publish, or preparation of the manuscript.

==============================
The Jurassic/Cretaceous (J/K) boundary, 145 million years ago, has long been recognised as an extinction event or faunal turnover for sauropod dinosaurs, with many ‘basal’ lineages disappearing. However, recently, a number of ‘extinct’ groups have been recognised in the Early Cretaceous, including diplodocids in Gondwana, and non-titanosauriform macronarians in Laurasia. Turiasauria, a clade of non-neosauropod eusauropods, was originally thought to have been restricted to the Late Jurassic of western Europe. However, its distribution has recently been extended to the Late Jurassic of Tanzania (Tendaguria tanzaniensis), as well as to the Early Cretaceous of the USA (Mierasaurus bobyoungi and Moabosaurus utahensis), demonstrating the survival of another ‘basal’ clade across the J/K boundary. Teeth from the Middle Jurassic–Early Cretaceous of western Europe and North Africa have also tentatively been attributed to turiasaurs, whilst recent phylogenetic analyses recovered Late Jurassic taxa from Argentina and China as further members of Turiasauria. Here, an anterior dorsal centrum and neural arch (both NHMUK 1871) from the Early Cretaceous Wealden Supergroup of the UK are described for the first time. NHMUK 1871 shares several synapomorphies with Turiasauria, especially the turiasaurs Moabosaurus and Tendaguria, including: (1) a strongly dorsoventrally compressed centrum; (2) the retention of prominent epipophyses; and (3) an extremely low, non-bifid neural spine. NHMUK 1871 therefore represents the first postcranial evidence for Turiasauria from European deposits of Early Cretaceous age. Although turiasaurs show clear heterodont dentition, only broad, characteristically ‘heart’-shaped teeth can currently be attributed to Turiasauria with confidence. As such, several putative turiasaur occurrences based on isolated teeth from Europe, as well as the Middle Jurassic and Early Cretaceous of Africa, cannot be confidently referred to Turiasauria. Unequivocal evidence for turiasaurs is therefore restricted to the late Middle Jurassic–Early Cretaceous of western Europe, the Late Jurassic of Tanzania, and the late Early Cretaceous of the USA, although remains from elsewhere might ultimately demonstrate that the group had a near-global distribution.

Introduction

The Late Jurassic is often regarded as a period of heightened sauropod dinosaur diversity, prior to a precipitous decline across the Jurassic/Cretaceous (J/K) boundary (145 million years ago), at which point many ‘basal’ sauropod lineages went extinct (Bakker, 1977; Hunt et al., 1994; Wilson & Sereno, 1998; Upchurch & Barrett, 2005; Barrett, McGowan & Page, 2009; Mannion et al., 2011). Increasingly, however, it is becoming apparent that any J/K extinction was not instantaneous (Tennant et al., 2017), at least for sauropods, with representatives of several ‘extinct’ sauropod groups now recognised from Early Cretaceous deposits (Gallina et al., 2014; Royo-Torres et al., 2014, 2017a, 2017b; Upchurch, Mannion & Taylor, 2015; D’Emic & Foster, 2016; McPhee et al., 2016).

The non-neosauropod eusauropod clade Turiasauria was first recognised by Royo-Torres, Cobos & Alcalá (2006) for three genera (Turiasaurus riodevensis, Losillasaurus giganteus, Galveosaurus herreroi) from the Late Jurassic of Spain (see Campos-Soto et al., 2017 regarding this revised age). Although Galveosaurus has subsequently been demonstrated to more likely represent a macronarian neosauropod (e.g. Barco, Canudo & Cuenca-Bescós, 2006; Carballido et al., 2011; D’Emic, 2012; Mannion et al., 2013), the western European record of named turiasaurs has since been expanded to include the Late Jurassic Portuguese taxon Zby atlanticus (Mateus, Mannion & Upchurch, 2014). In addition to postcranial remains, both Turiasaurus and Zby preserve teeth. These tooth crowns are mesiodistally broad relative to their apicobasal length, and have a distinctive ‘heart’-shaped outline (Royo-Torres, Cobos & Alcalá, 2006), narrowing mesiodistally along their apical halves (Mateus, Mannion & Upchurch, 2014). Primarily consisting of isolated teeth, additional remains have been referred to Turiasauria from contemporaneous Iberian deposits (Royo-Torres, Cobos & Alcalá, 2006; Royo-Torres et al., 2009; Mocho et al., 2016). Several authors have suggested that ‘heart’-shaped teeth from the Middle Jurassic–Early Cretaceous of the UK and France might also be attributable to turiasaurs, including the type specimens of ‘Cardiodon rugulosus’, ‘Neosodon’, and ‘Oplosaurus armatus’ (Royo-Torres, Cobos & Alcalá, 2006; Néraudeau et al., 2012; Royo-Torres & Upchurch, 2012; Mocho et al., 2016).

The distribution of turiasaurs was recently expanded to include the Early Cretaceous of the western USA (Royo-Torres et al., 2017a), based on relatively complete skeletons of two taxa, Mierasaurus bobyoungi (Royo-Torres et al., 2017a) and Moabosaurus utahensis (Britt et al., 2017), and thus confirming the group’s survival across the J/K boundary (Royo-Torres et al., 2017a). Finally, several remains from Africa have been suggested to represent turiasaurs. Mocho et al. (2016) commented upon similarities of two fragmentary Middle Jurassic teeth from Madagascar and Morocco, as well as a partial tooth from the Early Cretaceous of Libya, with European turiasaurs. Xing et al. (2015) also recovered the Middle Jurassic Moroccan sauropod Atlasaurus imelakei in a polytomy with Losillasaurus and Turiasaurus. Royo-Torres & Cobos (2009) suggested that several postcranial remains from the Late Jurassic Tendaguru Formation of Tanzania might also belong to Turiasauria. Most recently, Mannion et al. (in press) presented new anatomical data (see also Britt et al., 2017: 236) and phylogenetic analyses linking the enigmatic Tendaguru sauropod Tendaguria tanzaniensis with the turiasaur Moabosaurus. These authors recovered additional Late Jurassic taxa as possible turiasaurs: in some of their analyses, the Tendaguru sauropod Janenschia robusta and the Argentinean taxon Tehuelchesaurus benitezii were also placed in Turiasauria, whilst the Chinese sauropod Bellusaurus sui was consistently positioned as a turiasaur too.

Here, a previously undescribed anterior dorsal centrum and neural arch (NHMUK 1871) of a turiasaur from the Early Cretaceous Wealden Supergroup of the UK is presented. The putative turiasaurian affinities of several African and European occurrences are also discussed, including the utility of tooth morphology for identifying turiasaurs.

History and Provenance of NHMUK 1871

NHMUK 1871 is a relatively complete, but poorly preserved, anterior dorsal centrum and neural arch from an unknown Early Cretaceous ‘Wealden’ locality of the UK. Purchased by the NHMUK in 1891 as part of the Samuel H. Beckles collection, this specimen does not seem to have ever been mentioned in the published literature. Correspondence between Beckles and the NHMUK also does not provide any further information on the provenance of NHMUK 1871. Most of the dinosaur specimens collected by Beckles (e.g. the sauropod Haestasaurus (‘Pelorosaurus’) becklesii; Upchurch, Mannion & Taylor, 2015) came from the late Berriasian–Valanginian Hastings Group, in Hastings, East Sussex, southeastern England (Woodhams, 1990), and so this is the most likely source of NHMUK 1871. However, Beckles also collected material from elsewhere in the southeast of England, including the Isle of Wight (Woodhams, 1990), and so the specimen could conceivably have come from another Wealden locality. It also remains possible that NHMUK 1871 came from a slightly older stratigraphic unit, given that Beckles also collected fossil remains from the Berriasian section of the Purbeck Group (Owen, 1854), although the reported provenance of ‘Wealden’ suggests that this was probably not the case. Stratigraphically older and younger units in the areas in which Beckles collected were deposited under marine environments, and thus are also unlikely to have yielded NHMUK 1871. As such, although NHMUK 1871 is most likely to be late Berriasian–Valanginian, this cannot be conclusively demonstrated. Given the above discussion, it seems that the specimen can be attributed to the Wealden Supergroup, but it could conceivably have come from any section. Thus, the stratigraphic age of NHMUK 1871 can only be constrained to late Berriasian–early Aptian (Batten, 2011).

NHMUK 1871 comprises a centrum (including neural arch pedicels) and an unfused neural arch (Figs. 1 and 2). Although the two elements are a close match in size, it is not possible to re-articulate the centrum and neural arch, and this also results in an unusually dorsoventrally elongate neural canal. As such, it seems probable that they do not belong to the same vertebra. Both appear to be from the anterior region of the dorsal vertebral series though, and they probably represent approximately the second and third dorsal vertebrae. One further note of caution pertains to their preservation: whereas the neural arch is primarily black in colour, only a few small areas of the centrum display a similar colour. As such, although the available information indicates that they came from the same locality, and their relative sizes are consistent with being from the same individual, it is possible that the centrum and neural arch come from separate beds.

Figure 1 Photographs of the anterior dorsal centum NHMUK 1871.

(A) anterior, (B) posterior, (C) left lateral, (D) right lateral, (E) dorsal, and (F) ventral views. Abbreviations: ACDL, anterior centrodiapophyseal lamina; CPOL, centropostzygapophyseal lamina; CPRL, centroprezygapophyseal lamina; lpf, lateral pneumatic foramen; nc, neural canal; PCDL, posterior centrodiapophyseal lamina. Scale bar equals 100 mm. Photographs taken by the author.

Figure 2 Photographs of the anterior dorsal neural arch NHMUK 1871.

(A) anterior, (B) posterior, (C) right lateral, and (D) dorsal views. Abbreviations: dia, diapophysis; epi, epipophysis; PCDL, posterior centrodiapophyseal lamina; PODL, postzygodiapophyseal lamina; poz, postzygapophysis; prz, prezygapophysis; SDF, spinodiapophyseal fossa; SPOL, spinopostzygapophyseal lamina; SPRL, spinoprezygapophyseal lamina; TPOL, interpostzygapophyseal lamina; TPRL, interprezygapophyseal lamina. Scale bar equals 200 mm. Photographs taken by the author.

Systematic Palaeontology

Sauropoda Marsh, 1878

Eusauropoda Upchurch, 1995

Turiasauria Royo-Torres, Cobos & Alcalá, 2006

Turiasauria indet.

Material: NHMUK 1871, a relatively complete, but poorly preserved, anterior dorsal centrum (Fig. 1) and separate neural arch (Fig. 2).

Locality and stratigraphic position: Unknown locality, southeastern England, United Kingdom; probably from the Wealden Supergroup; late Berriasian–early Aptian (Early Cretaceous).

Description

The centrum is poorly preserved and incomplete, especially around the ventrolateral margins of its posterior cotyle (Fig. 1; see Table 1 for measurements). It is strongly opisthocoelous, and much wider mediolaterally than it is dorsoventrally tall (ratio = 1.44). The ventral surface is transversely convex, lacking ridges or excavations. Each lateral surface is too poorly preserved to determine whether the parapophyses were situated on the centrum or on the neural arch pedicels, although they are definitely absent from the preserved neural arch. Based on the right side of the centrum, a lateral pneumatic foramen is present (Fig. 1), but poor preservation and infilling by matrix mean that little of its morphology can be discerned. There is evidence for several poorly preserved laminae, comprising the anterior centrodiapophyseal lamina (ACDL), posterior centrodiapophyseal lamina (PCDL), centroprezygapophyseal lamina (CPRL), and centropostzygapophyseal lamina (CPOL) (Fig. 1). The neural arch pedicels terminate a short distance from the posterior margin of the centrum. The lack of fusion of both the centrum and neural arch with the rest of its respective vertebra indicates that this individual was not fully grown at the time of death.

Table 1 Measurements of the anterior dorsal vertebra NHMUK 1871.

Dimension	Measurement	
Centrum length (including condyle)	229	
Centrum length (excluding condyle)	170	
Anterior centrum dorsoventral height	158	
Anterior centrum mediolateral width	228	
Total preserved dorsoventral height of neural arch and spine	205	
Neural arch height	167	
Transverse width from midline to distal tip of right diapophysis	248	
Note:

All measurements in millimetres.

Erosion of the centrum in places reveals that it was pneumatised, with rounded camerae of ∼15 mm in diameter. No evidence for pneumaticity is visible in the neural arch. Unfortunately, attempts to CT scan the vertebra, to examine its internal tissue structure, were unsuccessful, as a result of its high density. As such, we cannot be sure whether the centrum was pneumatised by small camerae throughout, or if these were primarily restricted to near the outer bone surface.

In general, the neural arch is better preserved than the centrum (Fig. 2; see Table 1 for measurements). The flat articular surfaces of the widely separated prezygapophyses face dorsomedially and slightly anteriorly. They also expand anteroposteriorly towards their lateral tips. There is evidence for a V-shaped interprezygapophyseal lamina (TPRL), but this has been largely worn away (Fig. 2). The postzygapophyses are situated more dorsally than the prezygapophyses, and their articular surfaces face ventrolaterally and posteriorly. Overall, the zygapophyseal table is oriented at approximately 40° to the horizontal. There is no hyposphene, which is consistent with this being an anterior dorsal vertebra, and the postzygapophyses are connected by a horizontal interpostzygapophyseal lamina (TPOL). A prominent epipophysis is present on the dorsal surface of each postzygapophysis (Fig. 2).

The diapophyses project laterally and slightly ventrally, and there is evidence for a poorly preserved PCDL. The anterior and posterior surfaces of the diapophyses are unexcavated. A poorly preserved, near-horizontal postzygodiapophyseal lamina (PODL) is present. A shallow, dorsally-facing, elliptical spinodiapophyseal fossa (SDF) is situated anterior to the PODL, bounded anteriorly by the spinoprezygapophyseal lamina (SPRL) (Fig. 2).

Spinoprezygapophyseal laminae run dorsomedially from the middle of the posterior margin of the prezygapophyses. The anterior surface of the neural spine is transversely concave between the two SPRLs, and becomes rugose towards the midline, although there is no clearly defined prespinal ridge. The posterior surface of the neural spine is transversely concave, but poor preservation obscures whether a postspinal ridge or rugosity was present. Dorsomedially oriented, undivided spinopostzygapophyseal laminae (SPOLs) contribute to the posterolateral margins of the neural spine, but there are no spinodiapophyseal laminae (SPDLs). The dorsoventrally low, unbifurcated neural spine projects only very slightly above the level of the postzygapophyses, and is anteroposteriorly narrow, especially towards the midline (Fig. 2).

Discussion

Taxonomic affinities of NHMUK 1871

To determine the taxonomic affinities of NHMUK 1871, it is compared with anteriormost dorsal vertebrae from an array of eusauropods (see Fig. 3). A strongly dorsoventrally compressed centrum (mediolateral width to dorsoventral height ratio of >1.3) characterises the anterior dorsal vertebrae of several somphospondylan titanosauriforms (Mannion et al., 2013), the basal macronarian Lourinhasaurus (Mocho, Royo-Torres & Ortega, 2014), Apatosaurus (Gilmore, 1936), and Turiasauria (Royo-Torres et al., 2017a; Mannion et al., in press). The presence of camerae in the centrum is consistent with the anteriormost dorsal vertebrae of most eusauropods more derived than Omeisaurus, whereas the absence of clear camellae suggests that NHMUK 1871 lies outside of Titanosauriformes, and that it is not a mamenchisaurid (Wedel, 2003, 2005).

Figure 3 Comparative line drawings of anterior dorsal vertebrae of eusauropods.

Comparative line drawings showing dorsal vertebra two to three in anterior view for an array of eusauropods: (A) NHMUK 1871 (centrum + arch); (B) the turiasaur Tendaguria tanzaniensis (after Mannion et al., in press); (C) the turiasaur Moabosaurus utahensis (after Britt et al., 2017); (D) the mamenchisaurid Mamenchisaurus youngi (after Ouyang & Ye, 2002); (E) the diplodocid Apatosaurus louisae (after Gilmore, 1936); (F) the basal macronarian Camarasaurus supremus (after Osborn & Mook, 1921); (G) the brachiosaurid Europasaurus holgeri (after Carballido & Sander, 2014); and (H) the basal somphospondylan specimen known as the Cloverly titanosauriform (after D’Emic & Foreman, 2012). Vertebrae partially reconstructed where incomplete and not drawn to scale relative to one another.

The steeply inclined zygapophyseal table of NHMUK 1871 is most similar to the morphology in the anterior dorsal vertebrate of titanosaurs and rebbachisaurids, contrasting with the more shallowly oriented tables of other sauropods, including turiasaurs (Carballido et al., 2012; Poropat et al., 2016). Whereas the cervical vertebrae of most sauropods are characterised by the presence of epipophyses (Yates, 2007; Wilson & Upchurch, 2009; Mannion et al., 2013), their retention in anterior dorsal vertebrae is much less common, where they tend to be reduced structures (Mannion et al., in press). However, NHMUK 1871 shares the presence of prominent epipophyses with the turiasaurs Moabosaurus and Tendaguria, as well as Jobaria (Mannion et al., in press). Epipophyses are absent in other turiasaurs in which anteriormost dorsal vertebrae are preserved, that is, Mierasaurus and Turiasaurus (Royo-Torres, Cobos & Alcalá, 2006; Royo-Torres et al., 2017a; Mannion et al., in press). Only a small number of sauropod taxa are characterised by such a low neural spine in their anterior dorsal vertebrae, in which the spine is approximately level with the SPOLs. Euhelopus and Mamenchisaurus both share this feature, but the anterior dorsal neural spines of those taxa are bifid (Ouyang & Ye, 2002; Wilson & Upchurch, 2009). In contrast, the non-bifid anterior dorsal neural spines of the turiasaurs Moabosaurus (Britt et al., 2017), Tendaguria (Bonaparte, Heinrich & Wild, 2000) and, to a lesser extent, Mierasaurus (Royo-Torres et al., 2017a), strongly resemble that of NHMUK 1871. In contrast, the anteriormost dorsal vertebrae of Turiasaurus have dorsoventrally taller neural spines (Royo-Torres, Cobos & Alcalá, 2006; Royo-Torres et al., 2017a).

In summary, the combination of: (1) a dorsoventrally compressed centrum; (2) the retention of prominent epipophyses; (3) the low, non-bifid neural spine; and (4) the overall morphology of NHMUK 1871, more closely resembles the anteriormost dorsal vertebrae of turiasaurs than any other sauropods (Fig. 3). In particular, NHMUK 1871 appears to be most similar to Moabosaurus and Tendaguria. The anterior and posterior surfaces of the diapophyses of NHMUK 1871 are unexcavated though, contrasting with those two taxa (Mannion et al., in press). Despite the incomplete and fragmentary nature of NHMUK 1871, it appears to be readily referable to Turiasauria, more closely related to Moabosaurus + Tendaguria than to other turiasaurs.

Possible implications for turiasaurs from the Late Jurassic Tendaguru Formation of Tanzania

Upchurch, Mannion & Taylor (2015) recovered a sister taxon relationship between Janenschia and Haestasaurus (see also Mannion et al., in press), which are sympatric with Tendaguria and (probably) NHMUK 1871, respectively. Such close affinities might indicate a close faunal relationship between the latest Jurassic Tendaguru Formation and the Early Cretaceous Wealden Supergroup. Furthermore, this could conceivably be regarded as circumstantial evidence that Tendaguria is a junior synonym of Janenschia if NHMUK 1871 was recovered from the same area and stratigraphic bed as Haestasaurus. Given that both Janenschia and Tendaguria are recovered as turiasaurs in some of the phylogenetic analyses of Mannion et al. (in press), synonymy remains a possibility. However, until we find limb material associated with anterior dorsal vertebrae that can be referred to any of these taxa, such synonymisation cannot be justified.

Turiasaurian sauropod biogeography and evolutionary history

In addition to the named taxa Turiasaurus, Losillasaurus, Zby, Mierasaurus, Moabosaurus, and Tendaguria, several remains have been referred to Turiasauria (see Introduction). Most of these referrals are based on isolated teeth. Although the ‘heart’-shape is quite distinctive in most of the referred western European teeth, this is not the case in all instances (e.g. the type specimen of Oplosaurus armatus, from the Early Cretaceous of the UK), and especially not for the African specimens (two of which preserve only half of the crown). Mocho et al. (2016) identified three morphotypes of putative turiasaur teeth, which they suggested could be explained in two ways: either they represent different taxa, potentially including non-turiasaurs, or they are indicative of variation along the tooth row. The North American turiasaurs Mierasaurus and Moabosaurus show a clear heterodont dentition (Britt et al., 2017; Royo-Torres et al., 2017a), with subtle heterdonty present in Turiasaurus too (Royo-Torres & Upchurch, 2012). As such, the second hypothesis of Mocho et al. (2016) might well be correct. However, two of their morphotypes overlap with the teeth of other non-neosauropods (e.g. Jobaria; see also Mocho et al., 2016: fig. 7), and thus only broad, ‘heart’-shaped teeth can currently be attributed to Turiasauria with confidence. As such, the isolated teeth from the Middle Jurassic and Early Cretaceous of Africa cannot unambiguously be referred to Turiasauria, and are herein regarded as indeterminate eusauropods.

Xing et al. (2015) recovered the Middle Jurassic Moroccan sauropod Atlasaurus as a turiasaur in their phylogenetic analysis, but this result was not supported in recent studies that scored turiasaurian taxa based on firsthand observations (Mannion, Allain & Moine, 2017; Royo-Torres et al., 2017a; Mannion et al., in press). Very little published information is currently available for Atlasaurus, and it is in need of revision. As such, its phylogenetic affinities are uncertain (see Mannion et al., in press), but there is currently no evidence to support a turiasaurian placement. Finally, Mannion et al. (in press) recovered two Late Jurassic taxa within Turiasauria that would greatly extend the group’s distribution: the Argentinean sauropod Tehuelchesaurus, and the Chinese taxon Bellusaurus. However, those placements should be treated with caution: Tehuelchesaurus was placed outside of Turiasauria when extended implied weighting was applied, and Bellusaurus is known only from juvenile remains, which might affect its phylogenetic position (Moore et al., 2018). Furthermore, these positions have not been recovered in independent analyses (D’Emic, 2012; Royo-Torres & Upchurch, 2012; Carballido et al., 2017).

In summary, there is currently only unequivocal evidence for Turiasauria in the late Middle Jurassic–Early Cretaceous of western Europe (UK, France, Spain and Portugal), the Late Jurassic of Tanzania, and the late Early Cretaceous of the USA, but other remains suggest the possibility that the clade was more widespread, at least in the Late Jurassic.

Conclusions

A previously undescribed anterior dorsal centrum and neural arch (NHMUK 1871) from the Early Cretaceous Wealden Supergroup of the UK is recognised as a turiasaurian eusauropod dinosaur. This material shares several synapomorphies with Turiasauria, especially the Late Jurassic Tanzanian sauropod Tendaguria, and Moabosaurus, from the Early Cretaceous of the USA. NHMUK 1871 represents the first postcranial evidence for Turiasauria from European deposits of Early Cretaceous age. Unambiguous evidence for the non-neosauropod eusauropod clade Turiasauria is restricted to the late Middle Jurassic–Early Cretaceous of western Europe, the Late Jurassic of Tanzania, and the late Early Cretaceous of the USA, although remains from the Late Jurassic of Argentina and China might mean that the group had a near-global distribution.

I would like to express my gratitude to Paul Barrett and Susannah Maidment at the NHMUK for providing access to NHMUK 1871, as well as to all those who have facilitated the study of sauropod remains in their care. Comments from Rafael Royo-Torres also improved this manuscript.

Institutional Abbreviation

NHMUK Natural History Museum, London, UK.

Additional Information and Declarations

Competing Interests

Author Contributions

Data Availablity

The author declares that they have no competing interests.

Philip D. Mannion conceived and designed the experiments, performed the experiments, analyzed the data, contributed reagents/materials/analysis tools, prepared figures and/or tables, authored or reviewed drafts of the paper, approved the final draft.

The following information was supplied regarding data availability:

All data is provided in the article (main text and table of measurements).

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
