# Peer review of "A turiasaurian sauropod dinosaur from the Early Cretaceous Wealden Supergroup of the United Kingdom"

_PeerJ, doi:10.7717/peerj.6348_

## Round 0.1 · original submission · Minor Revisions

The manuscript requires only very minor revision (based on the comments of one reviewer) before it can be accepted for publication.

·

Basic reporting

The work is clear, well expressed and uses a scientific language. The article is written in English and uses a clear, unambiguous and technically correct text. The article conforms to professional standards of courtesy and expression.The article includes an adequate structure, with an introduction and sufficient background to demonstrate how the work fits the field of sauropod dinosaurs as well as an adequate paleontological systematic, description, discussion and conclusions. The literature is approved by adequate and sufficient references. The figures have quality and are relevant to show the results of the work, however a figure with more information would be convenient to help show the comparison and discussion between similar dorsal vertebrae in different groups of sauropods.

Experimental design

The research will be conducted rigorously and is based on previous works of high scientific quality about the investigated group of dinosaur sauropods. And the research keeps the ethical standards in force in its field. The methods that they expose allow it to be adequately followed by another researcher in order to reproduce it adequately in future works.
The work describes the presence of a clade of sauropod dinosaurs in UK that is producing novelties in recent years. In this sense, it is an isolated remain of a neural arch and a vertebral center assigned to a sauropod of the Turiasuria clade. This postcraneal remain would be the first time that it is cited in the United Kingdom, specifically a dorsal vertebra; but it is not the first time that this group is mentioned in the United Kingdom, which would not be a novelty, but it would provide new data and confirm the previous published data. Other authors have already mentioned the posible presence of this clade in the United Kingdom with teeth in the Middle, Upper and Lower Cretaceous of the United Kingdom (Royo-Torres & Upchurch, 2012; Mocho et al., 2016) (i.e. NHMUK R1610 from Wealden Group, Barremian-lower Aptian) and with some cladistic analyses for Haestasaurus (Mannion et al., in press). Also turiasaurs have been cited in the Lower Cretaceous of Europe, i,e, in base to heart-shaped teeth of Angeac (France) where the team of Ronan Allain have been found heart-shaped teeth (Neraudeau et al., 2012; Mocho et al., 2016) included in turiasaurs (Allain et al., 2016) from Berriasian sediments (Benoit et al., 2017).
If we take into account all the antecedents and they are accepted, the data presented here although with doubts in the age, would come to add new information about the presence of the turiasaurs in the Lower Cretaceous of Europe in a coherent way.

Allain, R. et al., 2016. Dinosaures, les géants du vignoble, Ediola editions, 248 p.
Benoit, R-A., Néraudeau, D., Martín-Closas, C. 2017. A review of the Late JurassiceEarly Cretaceous charophytes from the northern Aquitaine Basin in south-west France. Cretaceous Research, 79, 199-213.
Neraudeau et al. 2012. The Hauterivian-Barremian lignitic bone bed of Angeac (Charente, south-west France): stratigraphical, palaeobioloogical and paleogeographical implications. Cretaceous Research 37, 1-14
Mannion, P.D., Upchurch, P., Schwarz, D., Wings, O. In press. Taxonomic affinities of the putative titanosaurs from the Late Jurassic Tendaguru Formation of Tanzania: phylogenetic and biogeographic implications for eusauropod dinosaur evolution. Zoological Journal pf the Linnean Society.
Mocho, P., Royo-Torres, R. Ortega, F. Malafaia, E., Escaso, F. & Silva, B. 2016. Turiasauria-like teeth from the Upper Jurassic of the Lusitanian Basin, Portugal. Historical Biology, 28 (7), 861-880.
- Royo-Torres, R. & Upchurch, P. 2012. The cranial anatomy of the sauropod Turiasaurus riodevensis and implications for its phylogenetic relationships. Journal of Systematic Palaeontology. 10, 553-583.
- Upchurch, P., Mannion, P.D., Taylor, M.P. 2015. The anatomy and Phylogenetic Relationships of “pelorosaurus” becklesii (Neosauropoda, Macronaria) from the Early Cretaceous of England. Plos ONE 10(6): e0125819

Validity of the findings

The work is valued in a positive way, as it provides new data that can help confirm the presence of a clade of sauropod dinosaurs, the turiasaurs, in the Lower Cretaceous of Europe. However that age can not be accepted as safe. The uncertainty of not knowing the origin of the material and that comes from private collections without scientific information should put us in caution to draw conclusions in this regard. Geological information should be displayed with field data, stratigraphic columns and/or geological maps that support the geological age. However, this is seem impossible. Then, the material could be of older facies and belong to the sediments of the Purbeck of the United Kingdom and the material could be even from the Upper Jurassic. Can the author justify with more data the age he presents in his work?

The material is isolated and the comparison is with two taxa of the clade Turiasauria: Tendaguria and Moabosaurus. The first is also based on material isolated from the Tendaguru Formation, specifically 2 dorsal vertebrae. In both cases the results, although they may be considered valid, should be checked with more and new material in the future. The second taxon, with which it is compared, is Moabosaurus. It has enough cranial and postcranial remains to be assigned to Turiasauria with security and to be able to compare it with the anterior dorsal vertebrae of this work. But the author, it does not compare NHMUK 1871 directly with other dorsal vertebrae of the rest of turiasaurs such as Mierasaurus, Turiasaurus and Losillasaurus. At least it should be compared with the anterior dorsal vertebra of Mierasaurus to give more robustness to the work. In this sense to see the differences and similarities more clearly would be convenient and would help to make the work more interesting if a figure was added with the next information: illustrations of anterior dorsal vertebrae (first or second) for different groups of sauropods. In this way the author can discuss with reference to the figure why NHMUK 1871 is included in a specific group and excluded from others.

Additional comments

The work is very interesting, it provides new information about the group of turiasaurs. I think it should be published and make known to the scientific community the presence of this material as well as the comparison it makes with Tendaguria and Moabosaurus. This will undoubtedly help to know how the anterior dorsal vertebrae are in the Turiasauria clade and its comparison with other taxa. There is only a great doubt in the geological age of NHMUK 1871. This is due to the uncertainty of not being a material excavated by the author. In its history when proceeding from an old purchase by the NHMUK (in 1891) it lacks scientific methodology and the ignorance of its origin should make us doubt also about its age. I think this "uncertainty" should be highlighted in the title, abstract and conclusions to be taken into account in future work.

Reviewer 2 ·

Basic reporting

The manuscript presents a dorsal vertebra which morphology indicates affinities with the turiasaurian sauropod, extending the geographic and stratigraphic range of this lineage. It is a clear and well writen manuscript and I do not have any extra modification or suggestion for it

Experimental design

The designs are in agree with the material and the field

Validity of the findings

The arguments of the affinities are well demonstrated based on recent complete phylogenetic analysis

---

## Round 0.2 · accepted · Accept

The revised version of the manuscript fully addresses the reviewers' comments and will now be recommended for acceptance for publication.

#